# Tamoxifen Protects from Vesicular Stomatitis Virus Infection

**DOI:** 10.3390/ph12040142

**Published:** 2019-09-20

**Authors:** Lamin B. Cham, Sarah-Kim Friedrich, Tom Adomati, Hilal Bhat, Maximilian Schiller, Michael Bergerhausen, Thamer Hamdan, Fanghui Li, Yara Maria Machlah, Murtaza Ali, Vikas Duhan, Karl Sebastian Lang, Justa Friebus-Kardash, Judith Lang

**Affiliations:** 1Institute of Immunology, Medical Faculty, University of Duisburg-Essen, Hufelandstraße 55, 45147 Essen, Germany; laminbcham@gmail.com (L.B.C.); Sarah-KimFriedrich@gmx.net (S.-K.F.); Tom.Adomati@uk-essen.de (T.A.); bhathilal673@gmail.com (H.B.); Maximilian-Schiller@web.de (M.S.); michael.bergerhausen@gmail.com (M.B.); tamer_balaawi@yahoo.com (T.H.); fanghui.li@uni-due.de (F.L.); yaramachlah@yahoo.com (Y.M.M.); Murtaza142426@st.jmi.ac.in (M.A.); Duhan.Vikas@uk-essen.de (V.D.); KarlSebastian.Lang@uk-essen.de (K.S.L.); 2Department of Nephrology, University Hospital Essen, University Duisburg-Essen, Hufelandstraße 55 45147 Essen, Germany

**Keywords:** tamoxifen (TAM), vesicular stomatitis virus (VSV) infection, antiviral activity, interferon

## Abstract

**Background**: Tamoxifen (TAM) is an estrogen-receptor antagonist, widely used in the adjuvant treatment of early stage estrogen-sensitive breast cancer. Several studies have revealed new biological targets of TAM that mediate the estrogen receptor independent activities of the drug. Recently, the antiviral activity of TAM on replication of human immunodeficiency virus (HIV), hepatitis C virus (HCV) and Herpes simplex virus (HSV-1) in vitro was described. In the current study, we aimed to investigate the effect of TAM on infection with vesicular stomatitis virus (VSV). **Methods**: Vero cells were treated with different concentrations of TAM for 24 h and then infected with VSV. Additionally, C57BL/6 mice were pretreated with 4 mg TAM, one day and three days before infection with VSV. **Results**: Treatment of Vero cells with TAM suppressed the viral replication of VSV in vitro and in vivo. The inhibitory effect of TAM on VSV replication correlated with an enhanced interferon-I response and stimulation of macrophages. **Conclusions**: TAM was identified as being capable to protect from VSV infection in vitro and in vivo. Consequently, this antiviral function (as an advantageous side-effect of TAM) might give rise to new clinical applications, such as treatment of resistant virus infections, or serve as an add-on to standard antiviral therapy.

## 1. Introduction

Tamoxifen (TAM) belongs to the triphenylethylene class of molecules and acts as a selective estrogen receptor modulator (SERM) [1]. Through selective binding, it blocks the transcriptional activity of estrogen receptors and antagonizes the estrogen-mediated effects [2]. Estrogen receptors were discovered to be expressed on a wide range of cancers, especially breast cancer [3]. Approximately 70% of breast cancers were found to be positive for estrogen receptors alpha (ERα) and are consequently considered hormone-sensitive [4,5]. TAM exhibits an antitumor effect by estrogen receptor blockage and inhibition of proliferative signals driven by estrogens [6]. Therefore, TAM was approved more than 30 years ago as a targeted drug for the treatment of breast cancer [7]. Adjuvant therapy with TAM is a gold standard for the treatment of invasive breast cancer in early stages, as well as ductal carcinoma in situ [4,8]. TAM therapy was shown to improve survival and follow-up outcomes by preventing the recurrence of breast cancer after surgery, reducing the annual breast cancer death rate by 31% and the risk of developing a contralateral breast cancer by 50%, respectively [4,9,10,11]. However, recent expression of estrogen receptors beta was demonstrated for various solid tumors, encompassing carcinomas of the colon, pancreas, esophagus, stomach, brain, prostate, and lung [12,13]. This provides a rationale for the use of the estrogen-receptor antagonist properties of TAM as an applicable agent to treat tumors in other sites. In addition to the anti-estrogenic effects of TAM, recent experimental studies have described new biological effects of TAM on tumor cells that are independent of the estrogen receptor status [13,14]. The anti-proliferative activity of TAM on tumor cell differentiation and growth seems to be produced by direct interaction with protein kinase C [15,16]. Moreover, TAM was shown to inhibit other non-ER cellular targets, which are involved in signal pathways of tumor growth and development of resistance to chemotherapy [13]. Thus, treatment with TAM in vitro and in vivo leads to the inhibition of important processes that aid in tumor progression, such as angiogenesis, tumor invasion, and metastasis on the one hand; and helps to overcome multidrug resistance and induce programmed cell death on the other. The data underline the unique polyvalent nature of TAM as a targeted drug and support the idea of TAM as an effective therapy for estrogen-receptor-independent applications. TAM is already repurposed for the treatment of several estrogen-receptor-independent conditions, such as glioblastoma multiforme or desmoid tumors [17,18].

Beside the non-ER mediated effects on control of tumor growth, a broad anti-infective activity of TAM was discovered in the last few years. TAM acts against a wide range of microorganisms, including microbes, parasites, fungus, and viruses [19]. However, the exact mechanisms behind the inhibitory effect of TAM on infectious diseases are unknown; the non-ER mediated activities have been discussed. TAM was identified to have a strong antiviral activity and to disrupt viral replication in several in vitro studies for a number of viruses, such as HIV, HCV, HSV and Ebola [19]. Treatment with TAM had a suppressive effect on replication of HIV in infected lymphocytes, which was mediated through the mechanism of action independent of the estrogen receptor blocking the PCK and interfering with other targets of the NF-kappa B pathway [20]. TAM also acted against replication of HCV by preventing the activation of the RNA polymerase in an estrogen-receptor dependent manner and counteracting viral attachment, entry, replication, and exit of HCV, which appear to be related to the non-ER mediated effects of TAM [21,22,23]. Moreover, TAM elicits its antiviral activity in case infection with HSV-1 in vitro [24]. Hence, the application of TAM has inhibited viral fusion, cell penetration, and translocation of HSV-1 by blocking the chloride channel. The anti-infective effect of TAM was also registered in in vitro infection with several EBOV strains [25]. The aim of the current work was to explore the effect of TAM treatment on the course of infection with vesicular stomatitis virus (VSV).

## 2. Material and Methods

### 2.1. Mice and Virus

All the experiments were performed on mice housed in single ventilated cages. Animal experiments were carried out in accordance with the German law for animal protection and by authorization of the Veterinäramt Nordrhein Westfalen (Düsseldorf, Germany). C57BL/6 mice were purchased from Jackson Laboratory. Male and female C57BL/6J mice at least 8 weeks old with a minimal weight of 23–24 g were used. Ifnar^−/−^ mice have been previously described and were maintained on a C57BL/6 genetic background [26]. VSV, Indiana strain (VSV-IND, Mudd-Summers isolate), was originally obtained from Prof. D. Kolakofsky (University of Geneva, Geneva, Switzerland). The virus was propagated on baby hamster kidney 21 (BHK-21) cells at a multiplicity of infection (MOI) of 0.01, and then plaqued onto Vero cells [27]. The mice were infected intravenously with VSV (2 × 10^8^ PFU) for survival analysis, plaque assay, and histological analysis or with (2 × 10^6^ PFU) for the measurement of neutralizing antibodies. TAM (Sigma-Aldrich, St. Louis, MO, USA) was dissolved in corn oil, and 4 mg TAM (in 100 μL) per mouse was administrated intraperitoneally 3 days and 1 day before the VSV infection at indicated doses. 

For evaluation of TAM’s effect on virus replication in vitro, TAM at dosages of 7.5 µM, 15 µM, and 22.5 µM was added to the culture of Vero cells (provided by the Ontario Cancer Institute, Canada). TAM displayed a cytotoxic effect on Vero cells at a concentration of 50 µM. TAM was dissolved in DMSO, which served as a control for the in vitro experiments. After 24 h, the supernatant was removed and the cells were infected with VSV at the indicated concentrations. Virus titers were measured in the supernatant collected at different time points after the in vitro infection.

### 2.2. Plaque-Forming Assay

Virus titers of VSV were determined in organs and cell supernatant by plaque assay, as previously described [28]. Briefly, for quantification of the amount of VSV virus, cell supernatants or organs smashed in DMEM 2% FCS were titrated 1:3 over 12 steps and plaqued onto Vero cells (provided by the Ontario Cancer Institute, Canada). After 1 h of incubation at 37 °C, methylcellulose overly was added. After 24 h, the plaques were counted by crystal violet staining.

### 2.3. Histology

Histological analyses were conducted on snap-frozen liver and spleen tissues by using anti-VSV-G monoclonal antibody (clone Vi10, made in-house) as the primary antibody. Marginal zone macrophages were stained with PE conjugated anti-mSiglec1 antibody (R&D Systems, Minneapolis, MN, USA).

For H&E staining of spleen sections, slides with tissue sections were put in hematoxylin for 2 min and then in water for 10 min. The slides were then immersed in the differentiation solution 2–3 times, briefly dipped in 96% ethanol, and afterwards in eosin for 1 min. Thereafter, we immediately put the slides in 96%, 70% and 50% ethanol for 1 min, respectively and finally washed the slides in deionized water.

### 2.4. Neutralizing Antibody Assay

Neutralizing IgG antibodies were detected using a plaque reduction neutralization test. Serum samples were first pretreated with 0.1 M β-mercaptoethanol for 1 h to remove the IgM and IgA antibodies and then prediluted 1:40 with DMEM 2% FCS. The complement system was inactivated at 56 °C for 30 min. The serum was titrated 1:2 over 12 steps and incubated with 5 × 10^2^ PFU of VSV. After 90 min incubation at 37 °C, the mixture of virus and serum was plaqued onto the Vero cells. After 1 h, an overlay was added. The plaques were counted 24 h later by crystal violet staining.

### 2.5. ELISA

Interferon alpha levels were measured in murine serum by a commercially available ELISA assay (Thermo Fisher Scientific, Waltham, MA, USA) accordingly to the manufacturer’s instructions.

### 2.6. RT-PCR Analyses

RNA isolation, purification, and reverse-transcription to cDNA were performed as instructed by the manufacturer (Qiagen, Hilden, Germany). Quantification of RNA was done with a NanoDrop ND-1000 spectrophotometer (Peqlab Biotechnologie GmbH, Erlangen, Germany). For quantitative real-time PCR (qRT-PCR), Fast SYBR Green Master Mix (Applied Biosystems, Darmstadt, Germany) was used on the 7500 Fast Real-Time PCR System (Applied Biosystems, Darmstadt, Germany). Gene expression analysis was performed with GAPH, Ifnα2, Ifnα4, Ifnα5, Ifnβ1, Usp18, Irf7, Isg15, Mx1, Bst2, Oas1, Irf1, and Bcl2 assays (Qiagen, Hilden, Germany). For analysis, the expression levels of all target genes were adjusted to GAPDH expression levels (ΔCt). Gene expression values were then calculated based on the delta-delta-Ct (ΔΔCt) method relative to the naive controls. Relative quantities (RQs) were determined using the following equation: RQ = 2^−ΔΔCt^.

### 2.7. Flow Cytometry

CD8 positive cells were identified using PE-Cy7 labeled anti-CD8a antibody (eBioscience, Waltham, MA, USA).

To count CD8 + T cells producing INF-γ, spleens from infected mice were suspended into single cells, lysed with ammonium-chlorine-potassium (ACK), and restimulated at 37 °C with VSV-specific peptide p52 (PolyPeptide Laboratories, Strasbourg, France) for 4 h; brefeldin A (BFA) was added to it. Afterwards, the cells were stained for 30 min with surface markers, such as CD8a, fixed with 2% formaldehyde solution in PBS for 10 min, permeabilized with 1% saponin solution, and then stained with anti-IFN-γ antibody (XMG 1.1, eBioscience, Waltham, MA, USA) for 30 min at 4 °C.

## 3. Statistical Analysis

Data are presented as means with standard errors of the mean (SEM). Statistical significance between two groups was analyzed using unpaired two-tailed Student’s test. To compare the effects of different TAM concentrations in our in vitro studies, one-way ANOVA was carried out. Survival of the animals was determined by Kaplan-Meier survival analysis, and the log-rank (Mantel-Cox) test was used to test the significance of differences between the survival curves. P-values < 0.05 were considered statistically significant. Statistical analyses and graphical presentations were computed with GraphPad Prism Version 6 (GraphPad Software, Inc., La Jolla, CA, USA).

## 4. Results

### 4.1. TAM Inhibits Replication of vesicular stomatitis virus (VSV) In Vitro

We investigated the effect of treatment with TAM on viral replication in vitro. For this purpose, Vero cells were first pretreated with TAM, which was administrated in three different increasing concentrations. After 24 h, the TAM treated cells were infected with VSV. In case of VSV infection with MOI of 0.01, a significant reduction of virus load was observed only after 6 h after infection (compared to the untreated cells), while a significant decrease of VSV titers was seen after 4 h after VSV infection on using a higher MOI of 0.1 or 1 (Figure 1). The suppressive effect of TAM pretreatment on VSV replication occurred in a dose-dependent manner, which was clearly visible 24 h after infection. Consequently, TAM is able to inhibit viral replication of VSV in vitro.

### 4.2. TAM Pretreatment Protects from VSV Infection

Next, we questioned whether TAM may exhibit a similar inhibitory effect on viral replication in vivo. Therefore, C57BL/6 mice were treated twice with TAM 4 mg/100 µL 3 days and 1 day before the VSV infection, which was done with 2 × 10^8^ PFU on day 0. Immuno-histological staining of spleen sections harvested from the animals 8 h after VSV infection showed lower virus replication in mice pretreated with TAM than in the control mice (Figure 2A). Consistently, virus titers determined in spleen and liver tissues 8 h post infection were significantly reduced in TAM-treated mice, compared to the untreated controls (Figure 2B). Control mice pretreated with corn oil succumbed to the high-dose VSV infection, while mice which underwent TAM pretreatment showed less susceptibility to VSV and overcame the infection (Figure 2C). Next, we wondered whether TAM was also antiviral after the mice have been infected. For this therapeutic application, we first infected mice with VSV and then on days 2 and 3, treated them with TAM. This therapy improved the survival of treated mice, compared to the controls receiving only corn oil (Figure 2D).

### 4.3. TAM Pretreatment Reduces Antiviral Immune Response

Next, we aim to study antiviral immune responses in the presence of TAM. Surprisingly, TAM-treated mice had lower serum levels of total neutralizing and IgG neutralizing antibodies than the control mice (Figure 3A). Pretreatment with TAM resulted in a reduced total number of CD8^+^ T cells at day 10 after VSV infection relative to control mice (Figure 3B). Re-stimulation of the cells obtained from the spleen of TAM-pretreated mice with VSV-p52, a peptide derived from VSV, resulted in less activated interferon-γ producing CD8^+^ T cells in comparison to the control animals (Figure 3C). Collectively, pretreatment with TAM of C57BL/6 mice inhibits viral replication at an early time point in the case of VSV infection, but this effect seems to not be related to the presence of virus-specific cytokine-producing CD8^+^ T cells or increased production of virus-neutralizing antibodies.

### 4.4. Inhibitory Effect of TAM on Early VSV Replication Requires Interferon

To examine whether the inhibitory impact of TAM on early VSV amplification depends on innate type I interferon production, we measured the induction of interferon alpha and beta in the spleen 8 h after VSV infection. Induction of type I interferon was comparable between TAM pretreated and control mice (Figure 4A). In line with these data, we did not see any significant differences of interferon alpha serum levels between mice pretreated with TAM and the controls (Figure 4B). Interestingly, type I interferon-induced genes were differential regulated between TAM-pretreated and control mice (Figure 4C). While Isg15 and Mx1 were upregulated, Usp18, Oas1, and Bst2 seem to be insensitive to TAM (Figure 4C). To see whether these differences of interferon regulated genes could be the reason for different antiviral activity, we pretreated Ifnar^−/−^ mice with TAM and then infected them with VSV. In the absence of Ifnar, there was no antiviral activity of TAM (Figure 4D). Thus, Isg15 and Mx1, as well as other type I interferon stimulated genes were not affected by TAM in Ifnar^−/−^ mice (Figure 4E).

## 5. Discussion

In this study, we identified a protective role of TAM during VSV infection. Pretreatment with TAM leads to a strong inhibition of replication of VSV virus in vitro. The inhibitory effect of TAM is also persistent during VSV infection in vivo. Mice that were pretreated with TAM showed reduced quantities of VSV titers and survived longer after VSV infection.

Our observations on antiviral activity of TAM during VSV infection are in line with previous studies exploring the anti-infective ability of TAM on other viruses [19]. Pretreatment or treatment with TAM resulted in suppression of replication of HIV, HCV, and HSV-1 [19,20,21,22,23,24]. Consistent with our results, application of TAM prior to infection was more effective than TAM treatment after infection [19,20,21,22]. However, these data were derived from in vitro studies performed on cell culture systems infected with virus. In contrast, we showed an antiviral activity of TAM exposed not only in vitro, but also in vivo. We found that TAM-pretreated mice were more able to control VSV infection at an early time point—after 8 h—compared to the untreated controls. The suppressive effect of TAM on early VSV replication was most likely responsible for the improved survival of the pretreated animals after VSV infection. Mechanistically, we assume that the activation of macrophages and increased type I interferon response as important components of an innate immune system may contribute to a better control of VSV infection after TAM pretreatment. We saw the upregulated expression of Mx1 and Isg15 as interferon-induced genes occurring 8 h after VSV infection in TAM-pretreated mice. On the other hand, TAM lost its antiviral effect under the conditions of interferon-receptor deficiency, and the expression of interferon-induced genes was not influenced by TAM in mice lacking interferon receptors, providing evidence for our assumption. Additionally, recent studies demonstrated an induction of expression of interferon as well as interferon-regulated genes by TAM in breast cancer cells, supporting the proposed link between TAM treatment and enhanced interferon response [29,30]. Low titers of VSV-neutralizing antibodies, as well as a decreased number of CD8^+^ T cells in TAM-pretreated mice determined at the late phase (10 days) after VSV infection, suggest that the activation of adaptive immunity is unlikely linked to the antiviral function of TAM. TAM treatment seems to result in reduced production of antibodies by B cells and induce a decline of CD8^+^ T cells. These findings are in accordance with previous studies demonstrating an anti-proliferative effect of TAM on T and B cells [14,31]. Absolute numbers of lymphocytes were also reported to be reduced in breast cancer patients treated with TAM [32]. Alternatively, the early suppressive effect of TAM pretreatment on virus replication, which is responsible for reduced number of viral antigens, might lead to a weaker adaptive virus-specific response in mice pretreated with TAM, compared to the controls.

However, other potential mechanisms may be involved in the antiviral activity of TAM toward VSV infection. Previous reports have indicated that TAM targets multiple steps of the HCV viral life cycle [23]. TAM inhibited HCV replication by blocking ERα, which interferes with HCV-RNA polymerase NS5B and is needed for formation of HCV replication complex and promotion of HCV amplification (23). Furthermore, TAM appears to inhibit post-replication steps, such as assembly and release of HCV, and prevents the attachment of HCV on cell surface and virus entry [23]. The exact mechanisms of entry inhibition by TAM are not fully clear. Taking into account that TAM is a multitarget molecule, interacting with a number of targets such as protein kinase C, calmodulin or P-glycoprotein, it is conceivable that the antiviral effect of TAM might also be mediated through different other mechanisms independent of ER function [33]. Zheng et al. established an essential role of inhibition of chloride channel by TAM during infection with HSV-1 [24]. Blockages of chloride channel by TAM disrupted the fusion process of HSV-1 and limited HSV-1 replication [24]. Therefore, considering our results, we hypothesize that TAM might affect VSV through potential ER-independent mechanisms mentioned above. Especially in case of inhibition of VSV replication in Vero cells by TAM, we rely on effects independent of ER activity, because estrogen receptors are lacking on Vero cells [34]. Nevertheless, further studies are required to explore the detailed pathophysiological mechanisms underlying our observations of control of VSV infection after TAM pretreatment in vitro and in vivo.

In summary, we showed a significant inhibitory effect of TAM pretreatment on early VSV replication in vitro and in vivo, extending the existing knowledge of the antiviral properties of TAM. Usage of the off-target antiviral effects of TAM represents an attractive strategy to treat virus strains that are resistant to standard approved drugs. In addition, a combination of TAM with other antiviral agents might have a synergistic effect on viral replication and provide a promising tool to elevate the effectivity of anti-viral treatments. However, researches who are working with TAM in order to induce knock-outs of diverse genes, should be aware of the concomitant antiviral effect of TAM. This fact is highly relevant for the examination of the role of knock-out of different key mediators in viral replication and antiviral immune response because TAM pretreatment may mask stimulating effects on virus replication and change the corresponding phenotype.

Nevertheless, VSV predominantly affects animals. In humans, an occasional VSV infection can cause mild transient diseases; in particular, infections in animal-handlers or laboratory researchers were reported. Due to its favorable properties, VSV is suitable as a vaccine vector, as well as an attractive vector platform for viral-based immunotherapies. Over the last few years, the recombinant VSV-based vaccine against the Ebola virus, VSV-EBOV, was created, resulting in replication-competent VSV particles expressing the Ebola virus glycoprotein after replacement of VSV G from the virus genome [35]. The VSV G contributes to the neurotropic nature of VSV and the lack of the protein in the VSV-EBOV was shown to reduce the neurovirulence of the vaccine. Preclinical tests revealed the safety and efficacy of the VSV-EBOV with rapid protection potential against Ebola infection, enabling use in humans in ongoing clinical trials. In addition, the VSV vaccine platform is considered a promising candidate for adaptation for other pathogenic viruses with outbreak potential. Recombinant VSVs are also potent agents for application in oncolytic virotherapy. In a wide range of preclinical studies, VSV demonstrated oncolytic activity, enhancing the apoptosis of infected tumor cells and the ability to induce antitumor immune responses [36,37]. However, for high replication kinetics of the VSV occurrence of uncontrolled replication in patients, especially with compromised immune systems, receiving the VSV-based vaccines or oncolytic therapy might be conceivable, and in this context, TAM might serve as a treatment option to limit the progression of uncontrolled VSV replication and possible subsequent transmission.

## 6. Declarations

Ethic approval: The animal experiments were approved by the local ethics committee (proposal number A242).

## Figures and Tables

**Figure 1 pharmaceuticals-12-00142-f001:**
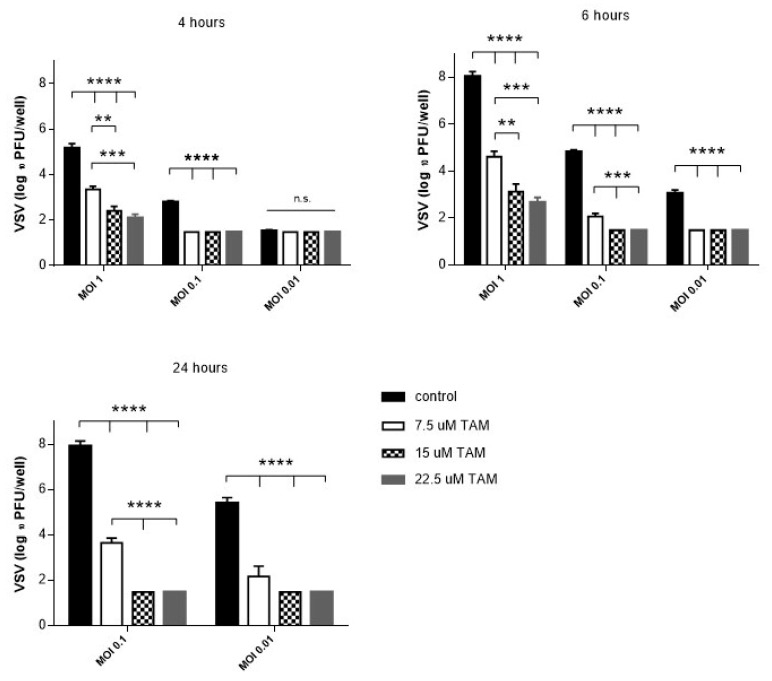
Tamosifen (TAM) pretreatment resulted in the inhibition of replication of vesicular stomatitis virus (VSV) in vitro. Vero cells were treated with 7.5 μM, 15 μM, 22.5 μM TAM, or 1 μL dimethyl sulfoxide (DMSO) used as a control for 24 h, and then infected with 0.01, 0.1, and 1 MOI of VSV for 1 h; the cells were washed with phosphate buffered saline (PBS) and fresh medium was added. VSV titers in cell supernatant were measured at 4 h, 6 h and 24 h after infection by plaque assay (*n* = 5). Data are expressed as means ± SEM. n.s.: not significant, ** *p* = 0.01; *** *p* = 0.001; **** *p* < 0.001.

**Figure 2 pharmaceuticals-12-00142-f002:**
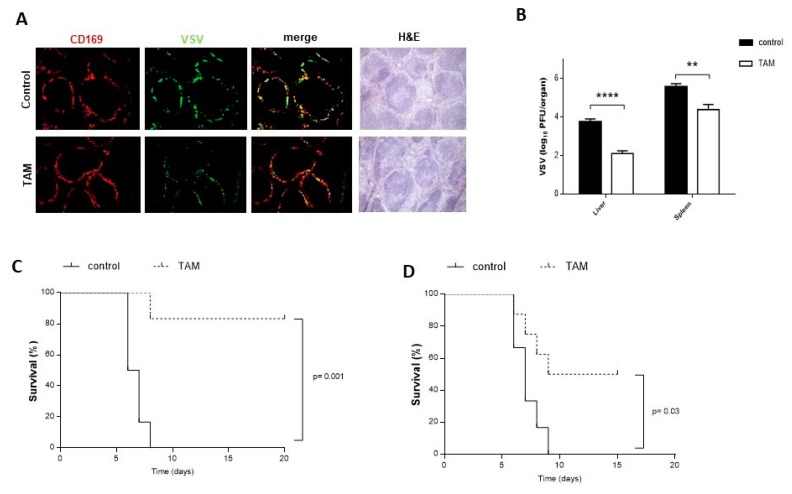
Pretreatment with TAM inhibits early VSV replication in vivo, improving survival after VSV infection. (**A**) Immunofluorescence and H&E staining of snap-frozen spleen tissues obtained from TAM pretreated and control mice 8 h after VSV infection. Spleen sections were stained for CD169 (red) and VSV glycoprotein (green). Scale bar = 100 μm; one representative out of 6 is shown. Fluorescent and light microscopy images were captured at 10x magnification using Keyence BZ-9000E microscope. (**B**) Virus titers were determined in liver and spleen tissues at 8 h post infection in TAM pretreated and control mice (*n* = 6). (**C**) C57BL/6 mice were pretreated intraperitoneally with 4 mg TAM at day -3 and day -1. Corn oil served as control. Survival was monitored in mice intravenously infected with 2 × 10^8^ PFU VSV at day 0 over the indicated period (*n* = 6). (**D**) Survival was monitored in C57BL/6 mice initially intravenously infected with 2 × 10^8^ PFU VSV at day 0 over the indicated period. TAM treatment (100 μL/4mg per mouse i.p.) was administrated twice on day 2 and 3 post VSV infection (*n* = 6 or 8). The error bars show SEM. ** *p* = 0.01; **** *p* < 0.001.

**Figure 3 pharmaceuticals-12-00142-f003:**
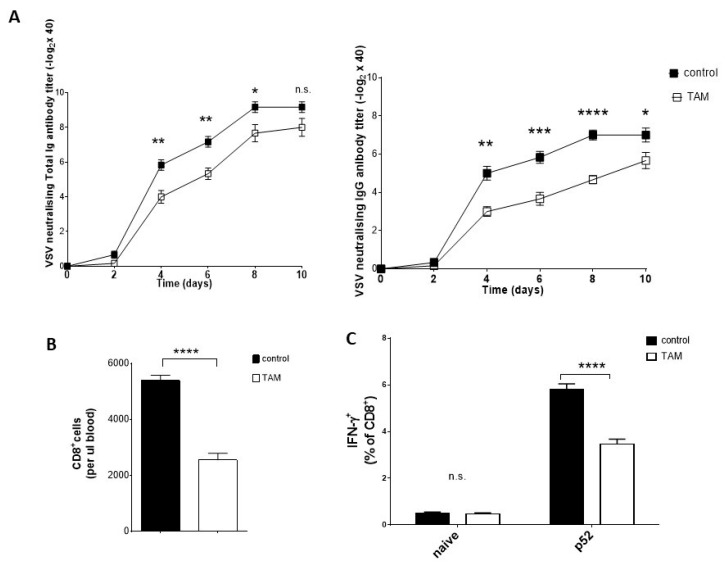
TAM suppresses the VSV neutralizing antibody response. (**A**) VSV neutralizing antibodies were measured in sera harvested from TAM pretreated C57BL/6 mice (4 mg TAM i.p. per mouse, applied at day -3 and -1) and control mice (treated with corm oil) at the indicated time points after infection with 2 × 10^4^ PFU VSV (*n* = 6). The left graph shows the total amount of VSV neutralising antibodies measured without pretreatment with β-mercaptoethanol. The right graph shows the titer of VSV neutralising IgG antibodies in serum that was pretreated with β-mercaptoethanol to remove IgM and IgA antibodies. (**B**) Total amount of CD8+ T cells was analyzed in blood of TAM-pretreated and control mice 10 days after VSV infection by flow cytometry (*n* = 6). (**C**) Percentage of CD8 + T cells from spleen capable to produce INF-γ after restimulation with virus-specific peptide p52 was measured in TAM pretreated and control mice 10 days after VSV infection by flow cytometry (*n* = 6). Data are expressed as means ± SEM. n.s.: not significant, * *p* = 0.05; ** *p* = 0.01; *** *p* = 0.001; **** *p* < 0.001.

**Figure 4 pharmaceuticals-12-00142-f004:**
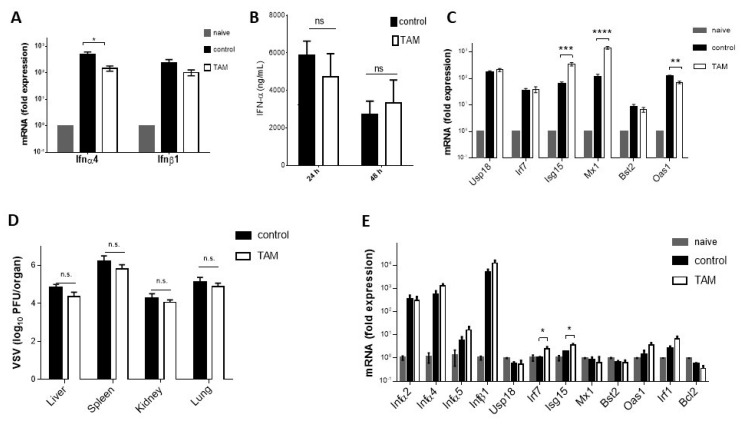
Inhibitory effect of TAM on early VSV replication requires interferon. (**A**) The mRNA expression of type I interferon from spleen tissue was determined by quantitative real-time PCR (qRT-PCR) 8 h after infection with 2 × 10^8^ PFU VSV in TAM pretreated and control mice (*n* = 5). (**B**) Interferon α concentrations were examined in sera of TAM-pretreated and control mice at 24 and 48 h after infection with 2 × 10^8^ PFU VSV (*n* = 6). (**C**) The mRNA expression of indicated interferon-induced genes from spleen tissue was determined by quantitative real-time PCR (qRT-PCR) 8 h after infection with 2 × 10^8^ PFU VSV in TAM-pretreated and control mice (*n* = 5). (**D**) Ifnar^−/−^ mice were pretreated intraperitoneally with 4 mg TAM or corn oil used as control at day -3 and -1, followed by VSV infection with 2 × 10^6^ PFU. Virus titers were determined in the liver, spleen, kidney, and lung of Ifnar deficient mice tissues 8 h post infection (*n* = 6). (**E**) The mRNA expression of indicated interferon-induced genes from spleen tissue was determined by quantitative real-time PCR (qRT-PCR) in Ifnar^−/−^ mice after treatment with 4 mg TAM at day -3 and -1 or corn oil used as control (*n* = 4). Spleen tissues were harvested 8 h post infection with 2 × 10^8^ PFU of VSV. Data are expressed as means ± SEM. n.s.: not significant, * *p* = 0.05; ** *p* = 0.01; *** *p* = 0.001; **** *p* < 0.001.

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
