# Peer review of "Tamoxifen Protects from Vesicular Stomatitis Virus Infection"

_pharmaceuticals, 2019, doi:10.3390/ph12040142_

Round 1

Reviewer 1 Report

In the manuscript “Tamoxifen protects from Vesicular Stomatitis Virus infection”, the authors examined anti-VSV effects of Tamoxifen, a widely used medicine for the treatment of early stage of estrogen-sensitive breast cancer. The authors found remarkable inhibition of VSV by Tamoxifen in vitro and in vivo.

This study has provided useful information on the antiviral effect of Tamoxifen against VSV replication, which may benefit the development of novel treatment of VSV infection. However, the data regarding to the mechanism of Tamoxifen’s antiviral effect is lacking.

Major concerns:

What’s the mechanism of anti VSV infection by Tamoxifen? The authors showed that Tamoxifen suppresses the VSV neutralizing antibody response, which is kind of confusing. It will be interesting to investigate which step of VSV replication cycle has been affected by Tamoxifen, e.g. early or late phase? How about the cytotoxicity of Tamoxifen in vitro and in vivo? Figure 1, the MOI for infection is pretty low (MOI 0.1 and 0.01), what’s the antiviral effect of Tamoxifen against higher MOI VSV infection, e.g. MOI 1 or above? Figure 2A, light microscopy and H&E staining of the spleen tissues (standard histology) should be included to better show the shape of spleen cells.

Minor concerns:

Figure 1, the Y-axis, it will be much clearer to use direct number in log scale, e.g. 1, 10, 100, 1000 etc. And in the lower segment of the Y-axis, the number is not clear, please revise. Figure 2b, the Y-axis, show the number directly. Figure 4E, in the Ifnar-/- mice, the treatment with TAM still showed significant difference for Isg15 as compared to the control. How to explain this data?

Author Response

Point by point reply

Reviewer: 1

In the manuscript “Tamoxifen protects from Vesicular Stomatitis Virus infection”, the authors examined anti-VSV effects of Tamoxifen, a widely used medicine for the treatment of early stage of estrogen-sensitive breast cancer. The authors found remarkable inhibition of VSV by Tamoxifen in vitro and in vivo.

This study has provided useful information on the antiviral effect of Tamoxifen against VSV replication, which may benefit the development of novel treatment of VSV infection. However, the data regarding to the mechanism of Tamoxifen’s antiviral effect is lacking.

Major concerns:

What’s the mechanism of anti VSV infection by Tamoxifen? The authors showed that Tamoxifen suppresses the VSV neutralizing antibody response, which is kind of confusing. It will be interesting to investigate which step of VSV replication cycle has been affected by Tamoxifen, e.g. early or late phase?

Answer to Reviewer 1 Comment 1:

With regard to the mechanism responsible for suppression of the VSV replication under tamoxifen pretreatment, we believe that enhanced interferon type I response induced by tamoxifen might contribute to the rapid control of the VSV-Infection after pretreatment with tamoxifen. We saw upregulation of expression of several interferon-induced genes 8 hours after VSV infection of C57BL/6 mice pretreated with tamoxifen compared to controls supporting our hypothesis. Additionally, the pretreatment with tamoxifen in Ifnar deficient mice did not limit the VSV infection 8 hours post infection and we did not observed any relevant differences in expression of interferon-induced genes compared to the untreated Ifnar mice.

As shown in Figure 3, titer of neutralising antibodies against VSV were decreased after tamoxifen pretreatment. This founding might be explained by the low number of viral antigens due to the early control of VSV replication after tamoxifen pretreatment that leads to reduced adaptive virus-specific response. Moreover, tamoxifen was reported to exert suppressive effects on proliferation of B and T-cells.

We observed a significant in vitro inhibition of VSV replication 4 hours after infection in Vero cells pretreated with tamoxifen. As previously reported, in case of HCV infection in vitro tamoxifen influenced viral attachment, entry, replication and exit of HCV. In our study, we focused on the potential mechanism of acting of tamoxifen in vivo during VSV infection and we found the association of tamoxifen treatment with increased interferon response. As already mentioned in the Discussion section, the investigation of further mechanism of antiviral acting of tamoxifen against VSV in particular the involvement of tamoxifen in diverse replication steps of the VSV might be objective of ongoing extensive in vitro studies that we cannot perform in the short time period (of 10 days given for revision).

How about the cytotoxicity of Tamoxifen in vitro and in vivo? Figure 1, the MOI for infection is pretty low (MOI 0.1 and 0.01), what’s the antiviral effect of Tamoxifen against higher MOI VSV infection, e.g. MOI 1 or above?

Answer to Reviewer 1 Comment 2:

Regarding the cytotoxicity of tamoxifen treatment in vitro, we observed cytotoxic effects on Vero cells in the concentration of 50µM occurring 48 hours after application of tamoxifen. The concentrations of tamoxifen used for our in vitro studies (7.5µM, 15µM and 22.5µM) did not provoke any morphological alterations of Vero cells 48 hours after tamoxifen treatment. We included the information on the concentration of tamoxifen inducing cytotoxicity into the Material and Methods section.

In terms of toxicity of tamoxifen in vivo, we administrated 4 mg of tamoxifen to the mice because it represents an eligible dose which did not cause any cytotoxicity or weight loss based on the experience of many research groups and therefore is widely used for example to activate the CRE in conditional knockout mice.  

Due to the fact, that VSV is a fast replicating virus having cytopathic effect on cultured cells lower doses of VSV are more appropriate to evaluate the replication pattern for later hours. Notwithstanding, as shown in the Figure I below tamoxifen also inhibits VSV replication in dose-dependent manner using the higher MOI of 1. But in vitro infection of Vero cells with 1 MOI of VSV results in an overwhelming VSV replication at later hours, e.g. 24 hours. As requested, we added the results on VSV infection in vitro with MOI 1 to the Results section and modified the Figure 1.

Figure I Tamoxifen pretreatment supresses replication of VSV in vitro using MOI 1. Vero cells were treated with 7.5 μM, 15 μM, 22.5 μM TAM tamoxifen or 1 μL DMSO used as control for 24 hours and then infected with 1 MOI of VSV for 1 hour, cells were washed with PBS and fresh medium was added. VSV titer in cell supernatant were measured at 4 hours and 6 hours after infection by plaque assay (n=6). Data are expressed as means ± SEM. n.s.: not significant, ** p= 0.01; *** p= 0.001; **** p< 0.001

Figure 2A, light microscopy and H&E staining of the spleen tissues (standard histology) should be included to better show the shape of spleen cells.

Answer to Reviewer 1 Comment 3:

As requested, we included the representative sections of snap-frozen spleen tissues using H&E staining into the Figure 2A.  However, we did not detect any impact of tamoxifen treatment on the architecture of spleen tissues after VSV infection. The protocol of the using H&E staining was added to the Material and Methods section.

Figure II H&E staining of snap-frozen spleen tissues obtained from tamoxifen pretreated and control mice 8 hours after VSV infection. Scale bar =100 μm; one representative out of 6 is shown. Light microscopy images were captured at 10x magnification using Keyence BZ-9000E microscope.

Minor concerns:

Figure 1, the Y-axis, it will be much clearer to use direct number in log scale, e.g. 1, 10, 100, 1000 etc. And in the lower segment of the Y-axis, the number is not clear, please revise. Figure 2b, the Y-axis, show the number directly.

Answer to Reviewer 1 Comment 4:

Thank you for the suggestions. We revised the Figure 1 accordingly to your recommendations.

To quantify the VSV titer we always use calculation in log10 PFU per well or organ and it represents a standard presentation of the virus titer.

Figure 4E, in the Ifnar-/- mice, the treatment with TAM still showed significant difference for Isg15 as compared to the control. How to explain this data?

Answer to Reviewer 1 Comment 5:

As shown in Figure 4E, tamoxifen pretreatment of the Ifnar-/- mice led to a slight upregulation of Isg15 compared to the control animals. However, it should be noted that in general the increase of expression of Isg15 was very low in comparison with naïve mice.  We suppose, that the observed raise of Isg15 mRNA expression in the Ifnar-/- mice might be attributed to the activation of Isg15 via interferon-independent pathways affected by tamoxifen treatment. Stimulation of a wide range of antiviral Isgs including the Isg15 independent of interferon was recently reported (Ashley CL, Abendroth A, McSharry BP, Slobedman B Interferon-independent upregulation of interferon-stimulated genes during human cytomegalovirus infection is dependent on IRF3 expression. Viruses 2019;11:pii: E246). 

Reviewer 2 Report

In this manuscript Cham et al. describe tamoxifen as VSV inhibiting drug. In the past tamoxifen has mainly been used for treatment of cancer. However, the substance has also been used to inhibit viruses in vitro, such as HIV HCV, and HSV-1. Here, the authors show that tamoxifen inhibits VSV in vitro and in vivo and describe an enhanced induction of the type I IFB response as potential mechanism.

Major comments:

The authors show that tamoxifen inhibits VSV replication and conclude that therefore a potential application for tamoxifen could be to treat virus infection in the clinic. However, VSV does rarely infect humans and also VSV infection of livestock is of low relevance. Here, testing tamoxifen on more clinically relevant infections would be interesting. However, VSV is used more and more as therapeutic agents (VSV-ZEBOV, oncolytic virotherapy). The authors could discuss tamoxifen as a treatment option for uncontrolled replication of these therapeutic VSV variants either in the patient itself or if the therapeutic virus is transmitted. There is only a small paragraph in the introduction on the current knowledge on the effects of tamoxifen on virus replication whereas the authors describe in length tamoxifen on tumor therapy. Given the topic of the manuscript, the focus should be more on tamoxifen and virus replication. Materials and methods: Please provide more information on mice (J or N C57BL/6, gender, age…); see also ARRIVE guidelines. Especially age is an interesting information to judge survival curves upon high dose VSV infection. Line 117: LCMV infection is mentioned, but no details and later no results for this are shown Materials and methods: description of intracellular cytokine staining is missing (see Figure 3C) Statistics: Please provide more details which student’s t test has been used. Why has a t test been used for comparison of more than two groups instead of ANOVA? Figures: Why are split axis used in some graphs (fig. 1, 2b, 3a, 4d)? Figures might be more easily understandable at the first glance without the split axis. Is log10 for the titers on the axis correct? Should it not be 10exp(x), e.g. for Fig1a a range of 10e2 to 10e8 PFU/well? Figure 1: Add axis labeling for each subgraph. Line 188: Why is DMSO used as control? Is tamoxifen for in vitro experiments dissolved in DMSO? Please add to methods. Lines 237-240 + 318-323: Authors claim that adaptive responses are not responsible for control of VSV in tamoxifen treated animals and that tamoxifen results in decreased induction of T and B cells. However, it seems more likely that the adaptive virus-specific responses are lower in the tamoxifen treated animals because of a reduced virus replication and consequently lower amount of viral antigens. Figure 3a: The authors claim that after b-ME treatment only IgG nab are measured as IgM nab are removed. Can other subtypes such as IgA be excluded? Left graph is without b-ME treatment and right with? This should be added to the legend. Line 257 says that IFN was analyzed in the spleen 24 and 48 hours after infection. However, the legend of fig 4 says 8 hours. In figure 4 different VSV doses for wt and IFNAR-/- mice were used, most likely as IFNAR-/- mice are more susceptible to VSV infection. However, this could influence the different induction of IFN-responsive genes in tamoxifen vs. control animals. Line 317: Why is activation of macrophages assumed to contribute to control of VSV after tamoxifen treatment?

Minor comments:

“ul” should be changed in “µl” Typo line 106: Vetirinäramt Typo line 222: 2 x 108 PFU Consistency: “Tamoxifen” (e.g. line 230) vs. “tamoxifen” (e.g. line 237) vs. “TAM” (figures)

Author Response

Point by point reply

Reviewer 2:

In this manuscript Cham et al. describe tamoxifen as VSV inhibiting drug. In the past tamoxifen has mainly been used for treatment of cancer. However, the substance has also been used to inhibit viruses in vitro, such as HIV HCV, and HSV-1. Here, the authors show that tamoxifen inhibits VSV in vitro and in vivo and describe an enhanced induction of the type I IFB response as potential mechanism.

Major comments:

The authors show that tamoxifen inhibits VSV replication and conclude that therefore a potential application for tamoxifen could be to treat virus infection in the clinic. However, VSV does rarely infect humans and also VSV infection of livestock is of low relevance. Here, testing tamoxifen on more clinically relevant infections would be interesting. However, VSV is used more and more as therapeutic agents (VSV-ZEBOV, oncolytic virotherapy). The authors could discuss tamoxifen as a treatment option for uncontrolled replication of these therapeutic VSV variants either in the patient itself or if the therapeutic virus is transmitted.

Answer to Reviewer 2 Comment 1:

Thank you for the comments. I agree that VSV is a zoonotic virus being mainly pathogen to animals and rarely leads to symptomatic disease in human working with the VSV livestock or infected animals. VSV is widely used for research purposes to study immunity, diverse virus mechanisms or antiviral effects of target molecules in mouse models. Thus, we used the VSV to study the antiviral effects of tamoxifen and some of its underlying mechanism in viral immunity. We presume that our knowledge of the antiviral effect of tamoxifen during in vivo infection with this cytolytic virus might be transferred to human-pathogen viruses belonging to the same family of Rhabdoviridae such as Rabies lyssavirus or to other human-pathogen cytolytic viruses. Unfortunately, the laboratory of our research group is an S2 not allowed us to work with human viruses. However, several previous reports described already antiviral activity of tamoxifen in in vitro infection with human-pathogen viruses such as HIV, HSV or HSV-1 suggesting tamoxifen to interfere with different steps of virus replication via estrogen-receptor independent pathways.

Thank you very much for the idea to discuss the possibility to use tamoxifen under conditions of potential uncontrolled replication in case of therapeutic application of VSV-based vaccines or oncolytic virotherapy in humans. We added the statement “Nevertheless, VSV predominantly affect animals. In humans an occasional VSV infection can cause mild transient disease, in particular infections of animal-handlers or laboratory researchers were reported. Due to its favorable properties VSV is suitable as a vaccine vector as well as   an attractive vector platform for viral-based immunotherapies. In the last years the recombinant VSV-based vaccine against the Ebola virus, VSV-EBOV, was created resulting in replication-competent VSV particles expressing the Ebola virus glycoprotein after replacement of VSV G from the virus genome (35). The VSV G contributes to the neurotropic nature of VSV and the lack of the protein in the VSV-EBOV was shown to reduce the neurovirulence of the vaccine. Preclinical tests revealed safety and efficacy of the VSV-EBOV with rapid protection potential against Ebola infection enabling use in humans in clinical trials which are already ongoing. In addition, VSV vaccine platform is considered as a promising candidate for adaptation for other pathogenic virus with outbreak potential. Otherwise, recombinant VSVs are potent agents for application for oncolytic virotherapy. In a wide range of preclinical studies VSV demonstrated oncolytic activity enhancing apoptosis of infected tumor cells and the ability to induce antitumor immune responses (36-37). However, regarding high replication kinetics of the VSV occurrence of uncontrolled replication in patients especially with compromised immune system receiving the VSV-based vaccines or oncolytic therapy might be conceivable and in this context TAM might serve as treatment option to limit the progress of uncontrolled VSV replication and possible subsequent transmission.” to the Discussion section.

There is only a small paragraph in the introduction on the current knowledge on the effects of tamoxifen on virus replication whereas the authors describe in length tamoxifen on tumor therapy. Given the topic of the manuscript, the focus should be more on tamoxifen and virus replication.

Answer to Reviewer 2 Comment 2:

As requested, we completed the Introduction section with information on the association of tamoxifen with viral infections.

The statement “TAM was identified to have a strong antiviral activity and to disrupt viral replication in several in vitro studies for a number of viruses such as HIV, HCV, HSV and Ebola (19). Treatment with TAM had a suppressive effect on replication of HIV in infected lymphocytes which was mediated through mechanism of action independent of the estrogen receptor blocking the PCK and interfering with other targets of the NF-kappa B pathway (20). TAM was also acting against replication of HCV by preventing the activation of the RNA polymerase in an estrogen-receptor dependent manner and counteracting viral attachment, entry, replication, and exit of HCV that appear to be related to the non-ER mediated effects of TAM (21-23). Moreover, TAM elicit its antiviral activity in case infection with HSV-1 in vitro (24).  Hence, application of TAM inhibited viral fusion, cell penetration, and translocation of HSV-1 by blocking chloride channel. Anti-infective effect of TAM was also registered in case in vitro infection with several EBOV strains (25).”  Was added to the Introduction section.

Materials and methods: Please provide more information on mice (J or N C57BL/6, gender, age…); see also ARRIVE guidelines. Especially age is an interesting information to judge survival curves upon high dose VSV infection.

Answer to Reviewer 2 Comment 3:

We used female and male C57BL/6J mice (stock number 000664) for our in vivo experiments who were at 8 weeks old and had a minimal weight of 23-24 g. As requested, we added this information to the Material and Methods section.

Line 117: LCMV infection is mentioned, but no details and later no results for this are shown.

Answer to Reviewer 2 Comment 4:

LCMV was omitted from the manuscript. We first also tested the effect of pretreatment with tamoxifen on the course of LCMV infection in vivo, but we decided not to include the data into the manuscript. We apologize for the oversight.

Materials and methods: description of intracellular cytokine staining is missing (see Figure 3C).

Answer to Reviewer 2 Comment 5:

As indicated in the figure legend for the Figure 3C, percentage of CD8 + T cells from spleen capable to produce INF –γ after restimulation with virus-specific peptide p52 was measured in tamoxifen pretreated and control mice 10 days after VSV infection by flow cytometry.  As requested, description of the intracellular cytokine staining “To determine the CD8 + T cells producing INF –γ , spleens from infected mice were suspend into single cells, lysed with ammonium-chlorine-potassium (ACK) and restimulated at 37 °C with VSV-specific peptide p52 (PolyPeptide Laboratories, Strasbourg, France) for 4 hours and brefeldin A (BFA) was added. Afterwards, cells were stained for 30 minutes with surface markers such as CD8a, fixed with 2% formaldehyde solution in PBS for 10 min, permeabilized with 1% saponin solution, and then stained with anti-IFN-γ antibody (XMG 1.1, eBioscience, Waltham, USA) for 30 minutes at 4 °C.” was added to the Material and Methods section.

Statistics: Please provide more details which student’s t test has been used. Why has a t test been used for comparison of more than two groups instead of ANOVA?

Answer to Reviewer 2 Comment 6:

We used unpaired two-tailed Student's test. As requested, we added the details on t test to the Material and Methods section. 

The data from the Figure 1 containing multiple groups with different concentrations of tamoxifen were analyzed using the one-way ANOVA test for multiple comparisons. We added this information to the Material and Methods section. For the other figures, e.g. Figure 2B or Figure 4, we considered the Student's test as an appropriate statistical test because we focused on comparison between TAM pretreated animals and the corresponding control group for virus titer in different organs or for expression of diverse interferon-induced genes. We assembled the virus titer in different organs as well as the mRNA expression of interferon type I and different interferon-induced genes in one graph instead of multiple separate graphs for each organ or gene for the purpose of clear arrangement of the results. But we were not interested in comparison of virus titer between several organs or of several interferon-induced genes using the ANOVA testing.

Figures: Why are split axis used in some graphs (fig. 1, 2b, 3a, 4d)? Figures might be more easily understandable at the first glance without the split axis. Is log10 for the titers on the axis correct? Should it not be 10exp(x), e.g. for Fig1a a range of 10e2 to 10e8 PFU/well? Figure 1: Add axis labeling for each subgraph.

Answer to Reviewer 2 Comment 7:

As requested, we changed format of the Y axis und revised all figures with the two segments Y axis.

The virus titer are calculated in log10 PFU per well or organ and it is a standard presentation of the VSV titer.

Figure 1 was further revised corresponding to your recommendations.

Line 188: Why is DMSO used as control? Is tamoxifen for in vitro experiments dissolved in DMSO? Please add to methods.

Answer to Reviewer 2 Comment 8:

DMSO was used as control for in vitro experiments because tamoxifen was dissolved in DMSO before application to the Vero cells. As requested, we included the statement “TAM was dissolved in DMSO, therefore DMSO served as control for in vitro experiments.”  into the Material and Methods section.

Lines 237-240 + 318-323: Authors claim that adaptive responses are not responsible for control of VSV in tamoxifen treated animals and that tamoxifen results in decreased induction of T and B cells. However, it seems more likely that the adaptive virus-specific responses are lower in the tamoxifen treated animals because of a reduced virus replication and consequently lower amount of viral antigens.

Answer to Reviewer 2 Comment 9:

We agree, that the reduced number of CD8+  T cells and lower amount of neutralizing antibodies in the group with tamoxifen pretreatment compared to the controls might also be explained by reduced response of the adaptive immune system to the lower amount of viral antigens  resulting of the rapid control of VSV replication at early stage due to pretreatment with tamoxifen. We added the statement “Alternatively, the early suppressive effect of TAM pretreatment on virus replication which is responsible for reduced number of viral antigens might lead to a weaker adaptive virus specific response in mice pretreated with TAM compared to the controls.” to the Discussion section.

However, we think that the previously described suppressive impact of tamoxifen on B and T cells as major components of the adaptive immune system should be taken into account and might play an additive role.

Figure 3a: The authors claim that after b-ME treatment only IgG nab are measured as IgM nab are removed. Can other subtypes such as IgA be excluded? Left graph is without b-ME treatment and right with? This should be added to the legend.

Answer to Reviewer 2 Comment 10:

β-Mercaptoethanol is well known to destroy the inter-chain sulfhydryl bond of gamma globulin. The pretreatment with β-mercaptoethanol is widely used to remove IgM as well as IgA antibodies and allows selective estimation of IgG antibody amount (Bansal F, Bhagat P, Srinivasan VK, Chhabra S, Gupta P. Immunoglobulin A gammopathy on serum electrophoresis: A diagnostic conundrum. Indian J Pathol Microbiol 2016;59:134-6). We added the information, that IgA antibodies were also removed from serum using pretreatment with β-mercaptoethanol to the Materials and Methods section.

The left graph of the Figure 3A demonstrates the total amount of VSV neutralising immunoglobulins wherefore pretreatment with β-mercaptoethanol was not required.  On contrary, the right graph shows titer of VSV neutralising IgG antibodies quantified after the appropriate pretreatment with β-mercaptoethanol. As requested, we indicated this information in the corresponding figure legend of Figure 3A.

Line 257 says that IFN was analyzed in the spleen 24 and 48 hours after infection. However, the legend of fig 4 says 8 hours.

Answer to Reviewer 2 Comment 11:

We apologize for having given incorrect time point of the measurements in the manuscript differing from the right time point indicated in the figure legend. In fact, the mRNA expression of interferon alpha and beta in the spleen was analyzed 8 hours after the VSV infection. However, serum levels of interferon alpha were measured 24 and 48 hours after infecting the mice with VSV. We corrected the mistake.

In figure 4 different VSV doses for wt and IFNAR-/- mice were used, most likely as IFNAR-/- mice are more susceptible to VSV infection. However, this could influence the different induction of IFN-responsive genes in tamoxifen vs. control animals.

Answer to Reviewer 2 Comment 12:

In the experiment performed to assess the mRNA expression of interferon and interferon-induced genes in spleen tissue of the Ifnar-/- mice 8 hours postinfection we used equal dosage of VSV (2×108 PFU) as for the wildtype mice in order to compare the effect of tamoxifen on interferon signaling between the wild type and the Ifnar-/- mice. The indication of the VSV dose of 2×108 PFU used in this separate experiment was added to the figure legend of Figure 4E.  

However, to determine the VSV titer in different tissues of the Ifnar-/- mice another experiment was performed and we choose lower dosage of VSV of 2 ×106 PFU due to the fast replication of VSV as indicated in figure legend of Figure 4D. 

Line 317: Why is activation of macrophages assumed to contribute to control of VSV after tamoxifen treatment?

Answer to Reviewer 2 Comment 13:

We mentioned in the discussion section that macrophages might additionally contribute to the beneficial effect of tamoxifen pretreatment on control of VSV infection. We previously performed an experiment that was not included into the manuscript but which might support the assumption (Figure III). C57BL/6J mice underwent depletion of macrophages on day -3 using intravenous injection of 300μL of clodronate. Then, tamoxifen was administrated to animals at day -3 and -1 followed by VSV infection (2×106 PFU VSV) conducted on day 0. 8 hours post VSV infection substantial VSV titer were detected in liver and spleen tissue harvested from clodronate treated mice as well as control mice (Figure III). Thus, in the absence of macrophages tamoxifen was not able to exhibit its antiviral activity.

Figure III Inhibitory effect of TAM on early VSV replication requires macrophages. C57BL/6 mice were first treated intravenously with 300μL clodronate at day -3 and afterwards 4 mg TAM or corn oil as control were applied intraperitoneally at day -3 and -1. Virus titer were measured in liver and spleen tissues harvested from the mice 8 hours after infection with 2×106 PFU VSV (n=6). Data are expressed as means ± SEM. n.s.: not significant.

On the other hand, there are several reports on the relationship between interferon type I and macrophages. It was shown that interferon alpha and beta are produced among others by macrophages, in particular by infected macrophages (Zhou Q, Lavorgna A, Bowman M, Hiscott J, Harhaj EW Aryl hydrocarbon receptor interacting protein targets IRF7 to suppress antiviral signaling and the induction of type I interferon. The Journal of Biological Chemistry 2015; 290: 14729–39). As previously observed, interferon alpha and beta can stimulate macrophages thereby affecting antiviral response (Vadiveloo PK, Vairo G, Hertzog P, Kola I, Hamilton JA Role of type I interferons during macrophage activation by lipopolysaccharide. Cytokine. 2000;12(11):1639-46; Zhou H, Zhao J, Stanley Perlman S Autocrine interferon priming in macrophages but not dendritic cells results in enhanced cytokine and chemokine production after coronavirus infection. MBio 2010;1. pii: e00219-10).

Minor comments:

“ul” should be changed in “µl” Typo line 106: Vetirinäramt Typo line 222: 2 x 108 PFU Consistency: “Tamoxifen” (e.g. line 230) vs. “tamoxifen” (e.g. line 237) vs. “TAM” (figures).

Answer to Reviewer 2 Comment 14:

We corrected the typing errors. As requested, we replaced the term “tamoxifen” by the abbreviation TAM throughout the manuscript.

Round 2

Reviewer 1 Report

The authors have answered my comments properly. I agree to accept this manuscript for publication at current version.